# Colorectal cancer with Synchronous liver-limited Metastases: the protocol of an Inception Cohort study (CoSMIC)

Ajith K Siriwardena,[1,2] Anthony K C Chan,[1,3] Agnieszka M Ignatowicz,[3] James M Mason,[3] On behalf of the CoSMIC study collaborators

## ABSTRACT

**Introduction** Colorectal cancer is the fourth most common cancer in the UK and an important cause of cancer-related death. In 20% of patients, there is metastasis to the liver or beyond at the time of diagnosis. The management of synchronous disease is complex. Conventional surgery removes the colorectal primary first, followed by chemotherapy, with resection of liver metastases as a final step. Advances in the availability and safety of liver surgery, anaesthesia and critical care have made two alternative options feasible. The first is synchronous resection of the primary and liver metastases. The second is resection of the metastatic disease as the first step, termed the reverse or liver-first approach. Currently, evidence is inadequate to inform the selection of care pathway for patients with colorectal cancer and synchronous liver-limited metastases. Specifically, optimal pathways are not defined and there is a dearth of prospectively recorded cohort-defining factors influencing treatment selection or outcome.

**Methods and analysis** Colorectal cancer with Synchronous liver-limited Metastases: an Inception Cohort (CoSMIC) is an inception cohort study of patients with a new diagnosis of colorectal cancer with synchronous liver-limited metastases. The sequence of treatment received, and factors influencing treatment decisions, will be evaluated against European Society of Medical Oncology guidelines. Clinical data will be collected, and quality of life, morbidity, mortality and long-term outcome compared for different treatment sequences adjusted for prognostic factors. Disease-free survival or progression will be measured at 1, 2 and 5 years. A nested qualitative study will ascertain patient experiences and clinician perspectives on delivery of care.

**Ethics and dissemination** The full study protocol was independently peer reviewed by Professor Kees de Jong (University of Maastricht, Holland). CoSMIC has ethical approval from the National Health Service Research Ethics Committee (14/NW/1397). Results will be disseminated to healthcare professionals and patient groups, and may be used to design a definitive trial addressing areas of equipoise in treatment pathways, as well as optimising current pathways to improve outcomes and experiences.

**Trial registration number** NCT02456285, pre-results.

## Strengths and limitations of this study

► This is the first prospective study directly comparing outcomes between the different surgical sequences of patients with colorectal cancer and liver-limited metastases.

► Completion of the Colorectal cancer with Synchronous liver-limited Metastases: the protocol of an Inception Cohort (CoSMIC) study protocol will provide important new evidence about the treatment of patients with colorectal cancer with synchronous liver-limited metastases and provide objective evidence to guide future studies (including randomised evaluations) in this area.

► The variables involved in the treatment allocation of such patients are vast and currently do not allow for a randomised controlled trial. The current CoSMIC study is therefore limited to an observational, inception cohort study.

[1]Regional Hepato-Pancreato-Biliary Unit, Manchester Royal Infirmary, Manchester, UK
[2]Faculty of Medicine, University of Manchester, Manchester, UK
[3]Warwick Medical School, University of Warwick, Coventry, UK

**Correspondence to**
Professor Ajith K Siriwardena;
ajith.siriwardena@cmft.nhs.uk

## BACKGROUND

Bowel cancer is the fourth most common cancer in the UK.[1] In Europe, colorectal cancer was the third most common cause of cancer and of cancer-related deaths in 2012.[2] The liver is the most frequent site of metastasis in colorectal cancer: 14%–20% of patients have hepatic metastases at presentation and up to a further third will subsequently develop liver lesions.[3 4] Liver metastases in patients with colorectal cancer are categorised as stage IV disease in which overall 5-year survival is 6%.[5] However, stage IV bowel cancer encompasses a wide clinical spectrum of disease ranging from patients with isolated hepatic metastases to patients with widespread metastatic disease. Patients with surgically resectable lesions confined to the liver have reported 5-year survival rates of 25%–40%.[6] Such patients represent a selected but important subgroup in whom long-term survival of approximately 17% at 10 years is feasible when the hepatic metastatic burden is removed.[7]

Patients who present with metastatic liver disease at a time point remote from their presentation with primary bowel cancer (termed metachronous disease) receive care focused on their new metastatic burden.[8 9] In

contrast, the management of patients who present with colorectal cancer and concurrent liver metastases (termed synchronous metastases) are more complex.[9 10] These patients may have less favourable cancer biology and thus may be less likely to become long-term survivors.[11] Logically, the management of patients with colorectal cancer with synchronous metastases can be dichotomised into those with hepatic disease together with extrahepatic metastatic disease and those with liver-limited metastatic disease. In the first category, systemic chemotherapy is the mainstay of treatment advocated in current guidelines for patients with advanced multisite metastatic disease of colorectal cancer origin.[9 12]

The second category of patients with liver-limited synchronous metastases represents a common and increasingly complex clinical management problem.[13] Traditional management (referred to variously as the classical or staged approach) comprised resection of the colorectal primary tumour followed by adjuvant chemotherapy with liver resection being undertaken (if at all) as a subsequent operation.[13–15] Key advances in the availability and safety of liver surgery, anaesthesia and critical care have made two alternative options feasible for patients with synchronous disease. The first is synchronous resection of the liver metastases and the colorectal primary.[13 15] This has the attraction of removing the macroscopic tumour burden with a single operation. However, the morbidity of complex liver resection combined with major bowel resection may be considerable and there is some evidence of a negative effect on progression-free survival.[16] The second option is resection of the liver metastatic disease as the first step, termed the reverse or liver-first approach.[17 18] Liver-first surgery may be particularly applicable to patients with rectal cancer with synchronous liver metastases where preoperative long-course chemoradiotherapy for the rectal primary prior to surgical resection creates a potential 'window' in which liver resection may be undertaken.[19 20] The liver-first strategy may also be oncologically advantageous by addressing the hepatic metastatic burden before progression in the liver renders this unresectable.[21] A further potentially important benefit of the liver-first approach is that pelvic surgery may be either avoided or less extensive in patients with rectal tumours with a complete endoscopic, radiological and clinical response to chemoradiotherapy.[22]

Currently, evidence is inadequate to inform the selection of care pathway for patients with colorectal cancer with synchronous liver-limited hepatic metastases. Specifically, there is a dearth of prospectively recorded cohort-defining factors influencing treatment selection or outcome. European Society of Medical Oncology (ESMO) guidelines[9] provide only a framework for the management of these patients. In the UK's National Health Service (NHS), treatment decisions are made at multidisciplinary cancer team (MDT) meetings that include liver surgeons, colorectal surgeons, oncologists and specialist nurses. Factors considered in formulating a treatment pathway include comorbidity and fitness for surgery, liver and colorectal disease distribution and the optimal placement of chemotherapy in the care plan.

Given the treatment permutations to be understood, an inception cohort study is valuable to understand patient outcomes as a function of clinical decisions and patient/disease characteristics. The Colorectal cancer with Synchronous liver-limited Metastases: an Inception Cohort (CoSMIC) study will recruit patients with colorectal cancer with synchronous liver-limited hepatic metastatic disease, and aims to fulfil four objectives. First, the study will characterise the management of this cohort by reporting relationships between modes of presentation, management and adherence to or deviation from current clinical guidelines. The second objective of the CoSMIC study is to provide (for the first time) comparable outcome data on patients with colorectal cancer with liver-limited hepatic metastases treated by synchronous or sequential surgery. The third objective is to address (also for the first time in a structured, prospective fashion) the impact of treatment on quality of life using validated questionnaire methodology. Finally, in this typically complex care plan, it may be difficult for the patients' voice to be heard and given due consideration. The focus on patient experience is important[23] and may vary substantially according to the treatment pathway. Thus, as a fourth objective a parallel qualitative study of both patient and clinician experience will help inform the knowledge of current practice. Completion of the CoSMIC study protocol will provide important new evidence about the treatment of patients with colorectal cancer with synchronous liver-limited metastases and provide objective evidence to guide future studies (including randomised evaluations) in this area.

## METHODS AND ANALYSIS
### Aims of the study
The primary aims of the study are to

1. characterise the management of patients with colorectal cancer and synchronous liver metastases, thus defining the relationship between presentation and treatment to demonstrate adherence to or deviation from an evidence-informed common pathway. It is accepted that modern management of this complex clinical scenario cannot be sufficiently addressed by a single pathway but the guidelines suggested by ESMO provide constrained management options: these include early use of neoadjuvant chemotherapy, surgical resection and adjuvant chemotherapy as the final stage. The treatment options within the common pathway standardise initial staging, accommodating treatment for liver metastases according to liver involvement and location of disease as well as different treatment requirements for patients with rectal primary cancer compared with those with colonic primary tumours;

2. provide comparable and prospective outcome data on patients with colorectal cancer with liver-limited hepatic metastases treated by synchronous or sequential surgery;

3. address the impact of treatment on quality of life using validated questionnaire methodology;

4. understand and explore the patient and caregiver experience of their disease and their experiences through the treatment pathway, including their voice in treatment decision planning;

5. understand and explore the clinician's experience of providing care to patients, and their perspectives on treatment pathways, specifically in areas of clinical equipoise, that make treatment allocation difficult;

6. explore the acceptability and barriers of a future randomised trial from both a clinician and patient's perspective, with a focus on the ethical dilemmas and the potential clinical value of such a study.

## Design

An inception cohort study will evaluate the treatment and outcomes of patients with colorectal cancer with synchronous liver-limited hepatic metastases. A parallel phenomenological qualitative study will also explore the patient and caregiver experience of the disease and treatment, and separately, the clinician perspective of providing care.

## Setting

The study population will comprise patients with colorectal cancer with liver-limited hepatic metastases referred to the Hepatobiliary Surgical Unit at Manchester Royal Infirmary—an NHS regional cancer-network approved hepato-pancreato-biliary (HPB) centre with a formally constituted and National Cancer Network peer-review accredited MDT. The study opened for recruitment in April 2015 with prospective recruitment to be undertaken for 24 months.

## Participants

To be eligible for inclusion in this cohort study, patients must fulfil the following:

### Inclusion criteria

1. Over 18 years of age
2. Able to give informed consent
3. Have a histological diagnosis of colorectal cancer
4. No prior history of malignancy.
5. Have radiological evidence on either contrast-enhanced CT or contrast-enhanced MR scanning of hepatic metastases at the time of diagnosis of the primary tumour or within 3 months thereof. Liver metastases should not be biopsied.
6. CT and/or fluorodeoxyglucose positron emission tomography ([18F]FDG-PET) evidence of the absence of extrahepatic metastases

7. MR scan assessment of local stage in those patients with rectal primary tumours
8. WHO performance status 0, 1 or 2 and considered by the MDT to be suitable for chemotherapy
9. A subset of patients from the cohort will be selected by purposeful sampling for the qualitative study following completion of their treatment, and are able to take part in a structured interview

### Exclusion criteria

1. Patients who are under 18 years of age
2. Patients who are unable to give informed consent
3. Patients who are unfit for the chemotherapy regimens in this protocol
4. Any psychiatric or neurological condition assessed by clinical judgement to compromise the patient's ability to give informed consent or to comply with oral medication
5. Partial or complete bowel obstruction not amenable to resolution by stent or diversion
6. Pre-existing neuropathy (> grade 1)
7. Patients with another previous or current malignant disease
8. Patients with known hypersensitivity reactions to any of the components of the study treatments
9. Patients with distant metastases outwith the liver
10. Patients who have received prior chemotherapy with oxaliplatin
11. Patients with a personal or family history suggestive of dihydropyrimidine dehydrogenase (DPD) deficiency or with known DPD deficiency
12. For patients selected for the qualitative study—those who are unable to give consent or are unfit to take part in a structured interview

For the clinician arm of the qualitative study, clinicians must fulfil the following:

### Inclusion criteria

1. Consultant grade
2. In clinical practice and an active participant of the HPB MDT in one of the following specialities: HPB surgery, colorectal surgery, radiology, oncology and histopathology
3. Willing to give informed consent

### Exclusion criteria

1. Non-consultant grade
2. Not in clinical practice or an active participant of the HPB MDT
3. Unwilling to give informed consent

## Recruitment

Patients will be formally identified prospectively at the weekly regional HPB MDT and approached in the outpatient clinic at MRI to discuss participation. Recruitment began in April 2015. For patients wishing to enrol, but where the treatment pathway has already started (typically

## Box 1    Baseline staging investigations and treatment details

**Baseline characteristics**
Patient demographics
Charlson comorbidity score
Blood tests (full blood count, serum urea and electrolytes, liver function tests, carcinoembryonic antigen)
**Clinical presentation**
**Cancer stage at presentation**
Location and stage (TNM/Dukes) of colorectal primary
Location, number and size of liver metastases
fluorodeoxyglucose positron emission tomography
Rectal MR (if applicable)
**Preoperative workup**
Portal vein embolisation
Cardiopulmonary exercise test
**Surgery (staged/synchronous resections)**
Sequence of surgery
Open/laparoscopic
Operative time
Estimated blood loss/transfusion
Bowel resection
Primary anastomosis
Covering stoma
**Liver resection**
Major resection (>3 Couinaud segments)
Pringle time
Complications (Clavien-Dindo)
Critical care stay
Total inpatient stay
Readmission within 30 days
30-day mortality
**Chemotherapy (neoadjuvant/adjuvant)**
Regime
Number of cycles (planned/given)
Duration of treatment
Side effects (common terminology criteria for adverse events)
Restaging
**Outcomes (1, 2 and 5 years)**
Disease-free survival
Disease progression
**Quality of life**
EuroQol with 5 dimensions with 3 levels of severity (EQ-5D-3L)
European Organization for Research and Treatment of Cancer Core Quality of Life Questionnaire (EORTC QLQ-C30)
European Organization for Research and Treatment of Cancer Colorectal Cancer Liver Metastasis specific module (EORTC QLQ LMC)

those who presented with an acute abdomen secondary to the bowel lesion), data will be retrospectively collected on treatment already received. Missing data points, particularly quality of life prior to surgery, during data analysis will be compensated for by unit imputation.

Potential participants of qualitative study will be approached following completion of their treatment either in the outpatient clinic or by telephone to ascertain their interest in taking part in the interviews.

### Data collection
Clinical data will be collected on the following baseline staging investigations and treatment details (box 1).

In addition to demographic detail, baseline staging will include tests for histological confirmation of cancer such as biopsy confirmation of a diagnosis of primary colorectal cancer (from the primary and not from the metastasis); tests for assessment of the liver and colorectal cancer in terms of lesion size, number, nodal involvement: contrast-enhanced CT scan and/or contrast-enhanced MR scan of the liver and pelvis and tests for assessment of the presence or absence of extrahepatic metastatic disease such as [18F]FDG-PET scan and serum assay of carcinoembryonic antigen. All of these tests are components of standard clinical care and no additional tests are undertaken for research purposes.

### Predictors of treatment allocation
Factors which guide clinical decision making in terms of the use of neoadjuvant chemotherapy and the choice of intervention (synchronous or sequential surgery).

### Timelines for completion of the treatment protocol
For the purposes of this study, this is defined as the amount of time in days from enrolment to completion of the protocol. The term 'protocol' relates to completion of the common treatment pathway.

### Failure to complete the treatment protocol
This is defined as dropout prior to completion of the allocated treatment sequence. It will be further categorised as due to disease progression, patient choice or unrelated to colorectal cancer (for example myocardial infarction) and will be recorded as the time in days from enrolment.

### Disease-free survival 12 months after enrolment into protocol
This is defined as the absence of tumour on a CT scan of the thorax, abdomen and pelvis undertaken at the completion of the protocol. In the case of those patients with rectal tumours treated by a' watch and wait 'policy, the term disease free can only be applied if there is a combination of radiological, endoscopic and clinical evidence of absence of cancer.

### Disease progression in patients who are not disease free at the end of protocol
The most sensitive measure of change is likely to involve a metric incorporating tumour size and number of lesions in the case of multiple metastases. There is evidence that CT-based volumetric assessment of metastases (seeded region growing method, slice-based segmentation or threshold-based segmentation) is more accurate for assessment of disease progression than the Response Evaluation Criteria In Solid Tumors (RECIST) V.1.1 method of largest axial diameter.[24] It is acknowledged that although RECIST criteria provide an objective means of assessment of solid tumour response to treatment, there is a risk of interobserver bias.[25] Further, RECIST criteria may be insufficient to assess response to treatment in patients with colorectal liver metastases treated

by biological agents such as bevacizumab.[26] Thus, disease progression at end of protocol will be assessed both by RECIST V.1.1 criteria and volumetric assessment.

### Resection margin status

The terms R0 bowel resection and R0 liver resection will be used (R0 means no tumour at or within 1 mm of surgical resection margin).[27 28]

### Complication and treatment-related morbidity profiles

Complications will be recorded prospectively according to the criteria defined above (see treatments) and assessed at the end of the study. Operative outcomes will be reported in keeping with the Dindo-Clavien system of assessment of postoperative morbidity.[29] The specific posthepatectomy complications of haemorrhage,[30] bile leakage[31] and liver failure[32] will be recorded in compliance with the guidance of the International Study Group of Liver Surgery. The morbidity associated with each intervention step will be recorded separately. Morbidity will include unplanned readmission and reoperation. Requirement for non-elective surgery for colonic complications (obstruction, perforation, bleeding) will be recorded.

### Mortality

Overall and cancer-related mortality in either arm after enrolment will be recorded. Mortality (and cause) will be determined using the Demographics Batch Service to access the national electronic database of the UK NHS.

### Use of stoma after colorectal surgery

Use of stoma (either temporary or permanent if this notification is available) will be recorded.

### Inpatient and critical care occupancy

A record will be made of inpatient and critical care occupancy associated with interventions; data will inform planning of economic evaluation in any subsequent randomised trial.

### Quality of life

Quality of life (QoL) will be assessed using the European Organisation for Research and Treatment of Cancer Core Quality of Life (EORTC QLQ-C30) and colorectal cancer liver metastasis specific module (EORTC QLQ-LMC) questionnaires, which have been validated for assessment of patient-reported outcomes during treatment of colorectal liver metastases.[33] The questionnaire will be completed by patients at time of enrolment and at 12 and 24 months. The EuroQoL questionnaire with 5 dimensions with 3 levels of severity (EQ-5D-3L)[34] will also be completed at the same time points, again supporting the design of future trial-based economic analyses.

### Qualitative study interview guide

Structured interviews will last approximately 45 min, and explore (1) for patients and caregivers: the experience of disease, particularly through the treatment pathway; understanding and expectations of time frame for investigations;

how they were informed of the diagnosis; how they received information related to the condition and treatment pathway; the type of information provided and who were the professionals explaining this; the nature and impact of information about diagnosis on patients and caregivers, and on their relationship with the clinician; aspects of the process patients and caregivers found useful/not so useful and what could be improved; and the acceptability of entering into a future randomised control trial; (2) for clinicians: their experience of providing care, in particular, the perspectives and their views on treatment pathways; difficulties and challenges around treatment allocation and decision-making processes; the relationship with patients; acceptability and barriers to entering patients who may be under their care into a future randomised trial; any ethical issues and the potential clinical value of future randomised control study.

### Data sources and measurements

Data will be collected prospectively using electronic study clinical case report forms. These will be anonymised and encrypted for storage and analysed prospectively during study to maximise data completion and resolve emergent problems in a timely fashion. The principal source of data will be the individual patient records. In addition, information will be gained by direct interview with patients (for quality of life assessment) and by interview with clinicians (for MDT choice decisions). Vital status beyond the duration of the study will be determined through the Demographics Batch Service of the NHS. Data will be reported at the end of year 3 allowing for a minimum 12 months outcome data in the entire cohort. It is also proposed (contingent on separate funding) that information on outcome will be collected for up to 5 years from study commencement, providing an informative survival analysis of treatment options.

Qualitative interviews will be audio recorded and transcribed verbatim.

### Study size

Based on clinical registers, the HPB unit at Manchester Royal Infirmary sees approximately 75 patients with colorectal cancer with synchronous liver-limited hepatic metastases per annum. As there are no study-related interventions, recruitment rates should be high and dropout low and is estimated to provide 150 patients in the 2-year recruitment period. A formal power calculation is not provided for this inception cohort study. Instead, the sample size is informed by the need to: provide stable estimates of variance for a range of outcomes; explore the relationship between the treatment pathway and health outcomes; estimate acceptability and recruitment rates; and describe patient and clinician experiences.

Purposeful sampling will be used to select patients from the cohort for the qualitative study. It is estimated that a sample size of four to six patients per group and one to two clinicians will produce data saturation. However, we will continue to interview until data saturation is reached.

## Analysis plan

The care of patients within the study pathway will be characterised by their principal treatment route as synchronous, liver first or bowel first. All patients will provide outcomes which will be included within analyses, grouped according to the treatment sequence received. Complication profiles in patients according to treatment group will be reported.

## Acknowledgement of selection bias

The liver metastases multidisciplinary team meeting at the Manchester Royal Infirmary is the sole forum approved by cancer commissioners for discussion of the care of patients with colorectal cancer liver metastases. The HPB unit guideline is that all patients with stage IV colorectal cancer should have their care reviewed at the MDT. However, it is acknowledged that there are several groups of patients who may bypass the MDT. In particular, patients with systemic disease 'beyond liver' may be referred for chemotherapy without consideration for liver surgery. From the patient's perspective, this care pathway is appropriate. Similarly, patients who present to local MDTs with liver metastases who undergo bowel-first surgery but whose disease progresses rendering them unsuitable for consideration for liver surgery will likely not be referred. For the purposes of reporting the CoSMIC data, these sources of patient loss to study will be acknowledged together with any potential for selection bias. Reporting will be pragmatic and descriptive.

## Statistical methods

Summary characteristics of patients, patient care provided and patient outcomes reported. Treatment centre characteristics will include measures of activity and surgical preference.

Exploratory analysis of process and clinical outcomes will be undertaken to explore the influence of patient, clinician, centre and treatment covariates, using regression modelling. Models will be subject to specification and robustness checks. Standard Generalised Linear Models (GLIM) and propensity score matching approaches will be compared with explore potential spectrum bias issues.

Interview transcripts will be managed by NVivo (QSR International, Melbourne, Australia) software. Interviews will be analysed thematically, using constant comparison[35] within a modified framework approach[36] coding both 'horizontally' (by coding each interview as a standalone hermeneutic unit) and 'vertically' (by scanning across the interviews for specific terms). Identified categories will be developed into a matrix of themes using mind-mapping techniques and a systematic cross-comparison will be undertaken to identify the similarities and differences between the different types of participant.

## Withdrawal from study

Patients will be able to withdraw from the study at any point. Data collected up to point of withdrawal will be retained for use within analyses.

## Quality control measures

Colorectal cancer cases and patients with liver metastases will have their care discussed at an appropriately constituted, UK cancer network-approved MDT.[37]

### Quality control in radiological images

Cross-sectional imaging will comply with the recommendations for cross-sectional imaging in cancer management of the Royal College of Radiologists.[38] An independent consultant radiologist will head the radiology standards group.

### Quality control in histopathological reporting

All histopathology reporting will be in compliance with the guidelines of the Royal College of Pathologists.[39]

## Health service cost of study

The clinical pathways within this study are cost neutral to the NHS as all the component steps are a part of current best practice. The study provides a structured template for progression through this pathway but all components are currently best standard care. Currently and in the near future, scientific and clinical equipoise are likely to be maintained. It will be possible to explore determinants of resource use within the common pathway as a study outcome.

## Adverse event reporting

Adverse events will be recorded, assessed for severity and attribution and reported in line with European Directive 2001/20/EC. In addition, if the Quality of Life assessment indicates that a patient is experiencing 'extreme problems' with their treatment, it would be an ethical duty of the CoSMIC research group to inform the clinical team involved with the care of the patient. This may introduce bias in subsequent quality of life assessments, and will be made transparent in any publication of results by the CoSMIC group.

Individual interviews will be stopped if there is any sign of emotional distress by either the patient or their relatives being interviewed. For any issues raised, with the patient's consent, we will contact their clinical team to make them aware of these issues so they can be formally addressed.

## DISSEMINATION POLICY

The results of CoSMIC will be presented at the appropriate conferences. Study outcome data will be set at 1, 2 and 5 years. Following publication of the final results, anonymised raw data will be made available.

**Collaborators** Aali J Sheen MD FRCS[1,2,3], Derek A O'Reilly PhD FRCS[1,2], Saurabh Jamdar MD FRCS[1,2], Rahul Deshpande MS FRCS[1], Nicola de Liguori Carino MD[1], Thomas Satyadas MS FRCS[1], Saifee Mullamitha MD MRCP[4], Michael Braun MRCP PhD[4], Nooreen Alam MRCP FRCR[4], Jurjees Hassan MSc MD FRCP[4], Gregory Wilson DRCOG FRCP Dip Onc[4], Thomas Treasure MD MS FRCS FRCP[5], Raj Rajashankar MRCP FRCR[6], Santhalingam Jegatheeswaran MRCS[1], Minas Baltatzis PhD[1], Professor Ray McMahon MD FRCPath[7], Rishi Sethi MRCS FRCR[6], James Hill ChM FRCS[8], David Smith BSc FRCS[9], Chris Smart FRCS[10], Arif Khan MS FRCS[11], Mohammud Kurrimboccus MD FRCS[12], Jonathan Epstein MD FRCS[13], Fergus

Reid FRCS[14], Kamran Siddiqui FRCS[15], Ramesh Aswatha MD FRCS[16] and Marius Paraoan MD FRCS[17]. [1]Regional Hepato-Pancreato-Biliary Unit, Manchester Royal Infirmary, [2]Faculty of Medicine, University of Manchester, [3]Manchester Metropolitan University, [4]Oncology Service, Christie Hospital, Manchester, [5]Clinical Operation Research Unit, University College London, [6]Department of Radiology, Manchester Royal Infirmary, [7]Department of Histopathology, Manchester Royal Infirmary, [8]Colorectal Surgery Unit, Manchester Royal Infirmary, [9]Department of Surgery, Royal Bolton Hospital, [10]Department of Surgery, Macclesfield District General Hospital, [11]Department of Surgery, Leighton Hospital, Crewe, [12]Department of Surgery, North Manchester General Hospital, [13]Department of Surgery, Salford Royal Hospital, [14]Department of Surgery, Stepping Hill Hospital, [15]Department of Surgery, Tameside General Hospital, [16]Department of Surgery, Wythenshawe Hospital, [17]Department of Surgery, Royal Albert Edward Infirmary.

**Contributors** AKS and JMM conceived the study concept. All the authors developed and modified the study design and protocol. AKS is the chief investigator of the study and takes overall responsibility for all aspects of the study design and trial conduct. All authors and collaborators have read and approved the final manuscript.

**Funding** This work was supported by a grant from the Dickinson Trust of the Manchester Royal Infirmary (received by AKCC).

**Competing interests** None declared.

**Ethics approval** National Research Ethics Service North West Committee.

**Provenance and peer review** Not commissioned; externally peer reviewed.

**Data sharing statement** Population-based data will be shared on request, although confidentiality issues will limit the sharing of individual-level data for this paper.

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
