## [Reviewer comments · BMJ Open]

ARTICLE DETAILS

TITLE (PROVISIONAL)	Colorectal cancer with Synchronous liver-limited Metastases: The protocol of an Inception Cohort study (CoSMIC).
AUTHORS	Siriwardena, Ajith; Chan, Anthony; Ignatowicz, Agnieszka; Mason, James

VERSION 1 - REVIEW

REVIEWER	Stephen J Wigmore Professor of Transplantation Surgery University of Edinburgh, UK
REVIEW RETURNED	17-Nov-2016

GENERAL COMMENTS	This manuscript represents the study protocol for the "CoSMIC" study which is an inception cohort study for patients with colorectal cancer who present with synchronous liver metastases but no other disease. The management of these patients is difficult and variable and the authors are correct in their assertion that there are no universally accepted guidelines for treatment order. This is even more the case since the EPOC study really failed to make a strong case for chemotherapy before liver surgery. The protocol is well written and presents what is a pragmatic approach to studying a question which is unlikely to be addressed by a conventional randomised controlled trial. Comments 1. This study is inevitably going to be prone to significant selection bias. Ifs the MDT in Manchester is anything like my own MDT then it is likely that patients who are offered synchronous liver and bowel surgery will be patients in whom there is threat of bowel obstruction and relatively easy liver resection or right sided colonic disease with an opportunity for atypical or 2-3 segment resection of the liver. In the same way the patients who traditionally go down a liver first approach are those that are most at risk of becoming non-operable with clear margins based on their liver disease. This issue of selection bias may not be such an issue because of the design of the study but may have an influence on outcomes for patients. One approach to trying to overcome this would be to have an external virtual MDT where cases are reviewed to assess whether they genuinely could have had any of the approaches allowed in the study or whether they would always be selected for one protocol over all others. 2. The justification for the liver-first strategy "may also be oncologically advantageous if liver metastatic disease rather than the primary cancer gives rise to systemic metastasis – although this is not fully established". I think this is quite a weak argument and probably not
--

	supported by evidence. The risk is usually in the liver metastases becoming un-resectable with clear margins rather than them giving rise to new metastases. 3. On a philosophical level it seems rather odd to review and publish a protocol for a study which began recruiting around 18 months ago and should only have 6 months more to complete. There is no real opportunity to revise the protocol and so it is what it is to a certain extent.
--	--

REVIEWER	Shinichiro Takahashi National Cancer Center Hospital East, JAPAN
REVIEW RETURNED	01-Dec-2016

GENERAL COMMENTS	This study aims to compare various outcomes according to the sequence of primary resection, hepatectomy, and chemotherapy in pts with synchronous colorectal liver metastases. There is little evidence in this field. Then, this cohort study seems interesting for HBP and colorectal surgeons, but looks also challenging. There are some problems to be clarified. 1. This is a single-institutional study. There may be standard criteria to decide the treatment sequence for pts with synchronous metastases in their institution. For instance, when tumor burden in liver is large, liver-first strategy might be chosen. It should be stated. 2. The purpose of the study is somewhat vague. Is the main purpose of the study investigation of spread, backgrounds and outcomes of each strategy? Or does this study aim to clarify better strategies in some point of view among 3 strategies by comparing their outcomes using propensity score? If the former is true, there does not seem to be a definitive advantage in this cohort study comparing with retrospective one in spite of numerous efforts. If the latter, this study looks more attractive but there must be a statement about statistical methods, endpoints, and appropriate hypothesis to evaluate and compare three strategies. The authors should state how to utilize the results for pts with synchronous colorectal hepatic metastases.
--

VERSION 1 – AUTHOR RESPONSE

REVIEWER 1

Point 1:

This study is inevitably going to be prone to significant selection bias. If the MDT in Manchester is anything like my own MDT then it is likely that patients who are offered synchronous liver and bowel surgery will be patients in whom there is a threat of bowel obstruction and relatively easy liver resection or right-sided colonic disease with an opportunity for atypical or 2-3 segment resection of the liver. In the same way the patients who traditionally go down a liver-first approach are those that are most at risk of becoming non-operable with clear margins based on their liver disease. This issue of selection bias may not be such an issue because of the design of the study but may have an influence on outcomes for patients. One approach to trying to overcome this would be to have an external virtual MDT where cases are reviewed to assess whether they genuinely could have had any of the approaches allowed in the study or whether they would always be selected for one protocol over all others.

Response to point 1:

Reviewer 1 is correct in that there is likely to be selection bias in recruitment here. This is an important point which we acknowledge. To address this, we have added an additional section in the protocol entitled "Acknowledgement of selection bias".

We acknowledge this issue (which is common to studies which report patients referred to liver units for liver surgery) by the following:

The liver metastases multidisciplinary team meeting at the Manchester Royal Infirmary is the sole forum approved by cancer commissioners for discussion of the care of patients with colorectal cancer liver metastases. The HPB unit guideline is that all patient with stage IV colorectal cancer should have their care reviewed at the MDT. However, it is acknowledged that there are several groups of patients who may bypass the MDT. In particular, patients with systemic disease "beyond liver" may be referred for chemotherapy without consideration for liver surgery. From the patient's perspective, this care pathway is appropriate. Similarly, patients who present to local MDTs with liver metastases who undergo bowel-first surgery but whose disease progresses rendering them unsuitable for consideration for liver surgery will likely not be referred. For the purposes of reporting the CoSMIC data these sources of patient loss to study will be acknowledged together with any potential for selection bias. Reporting will be pragmatic and descriptive.

Point 2:

The justification for the liver-first strategy "may also be oncologically advantageous if liver metastatic disease rather than the primary cancer gives rise to systemic metastasis – although this is not fully established". I think this is quite a weak argument and probably not supported by evidence. The risk is usually in the liver metastases becoming unresectable with clear margins rather than them giving rise to new metastases.

Response to Point 2:

This point is also accepted and the text modified. The statement now reads (Page 5 – first paragraph): The liver-first strategy may also be oncologically advantageous by addressing the hepatic metastatic burden before progression in the liver renders this unresectable

Point 3:

On a philosophical level it seems rather odd to review and publish a protocol for a study which began recruiting around 18 months ago and should only have 6 months more to complete. There is no real opportunity to revise the protocol and so it is what it is to a certain extent.

Response to point 3:

This point is interesting and is correct. It is indeed correct that the current intention is to close the study to recruitment in April 2017. Worldwide there are several groups studying patients with colorectal cancer and synchronous metastases and the value of reporting the CoSMIC protocol is that it allows other groups to compare and contrast their study concepts with ours. There is also a very real sense in that (as reviewer 1 states earlier) that a simplistic randomized trial cannot be conducted in this setting and some of the features highlighted in the CoSMIC protocol may be of value to others looking to design future studies.

Reviewer: 2

This study is aim to compare various outcomes according to the sequence of primary resection, hepatectomy, and chemotherapy in pts with synchronous colorectal liver metastases. There is little evidence in this field. Then, this cohort study seems interesting for HBP and colorectal surgeons, but looks also challenging. There are some problems to be clarified.

Point 1:

This is a single-institutional study. There may be standard criteria to decide the treatment sequence

for pts with synchronous metastases in their institution. For instance, when tumor burden in liver is large, liver-first strategy might be chosen. It should be stated.

Response to Point 1:

This is a very important point. We have now added text explaining the rationale for treatment selection (Page 7) as follows: It is accepted that modern management of this complex clinical scenario cannot be sufficiently addressed by a single pathway but the guidelines suggested by ESMO provide constrained management options: these include early use of neo-adjuvant chemotherapy, surgical resection and adjuvant chemotherapy as the final stage. The treatment options within the common pathway standardise initial staging, accommodating treatment for liver metastases according to liver involvement and location of disease as well as different treatment requirements for patients with rectal primary cancer compared to those with colonic primary tumours

Point 2:

The purpose of the study is somewhat vague. Is main purpose of the study investigation of spread, backgrounds and outcomes of each strategies? Or does this study aim to clarify better strategies in some point of view among 3 strategies by comparing their outcomes using propensity score? If the former is true, there does not seem to be definitive advantage in this cohort study comparing with retrospective one in spite of numerous efforts. If the latter, this study looks more attractive but there must be statement about statistical methods, endpoints, and appropriate hypothesis to evaluate and compare three strategies. The authors should state how to utilize the results for pts with synchronous colorectal hepatic metastases

Response to Point 2:

This is also an important point. Potential readers of the protocol and the future final report should be able to understand the goals of the study.

We have enhanced the statistical methods section and provide the following text (Page 17) :

Summary characteristics of patients, patient care provided and patient outcomes reported. Treatment centre characteristics will include measures of activity and surgical preference.

Exploratory analysis of process and clinical outcomes will be undertaken to explore the influence of patient, clinician, centre and treatment covariates, using regression modelling. Models will be subject to specification and robustness checks.

VERSION 2 – REVIEW

REVIEWER	Shinichiro Takahashi National Cancer Center Hospital East, Japan
REVIEW RETURNED	22-Jan-2017

GENERAL COMMENTS	Unfortunately, no revision was made for some of the points raised by reviewers. However, it is understandable that revision of the protocol of the study which started more than 2 years before is difficult.
---